# SARS-CoV-2 Spike Protein Binding of Glycated Serum Albumin—Its Potential Role in the Pathogenesis of the COVID-19 Clinical Syndromes and Bias towards Individuals with Pre-Diabetes/Type 2 Diabetes and Metabolic Diseases

**DOI:** 10.3390/ijms23084126

**Published:** 2022-04-08

**Authors:** Jason Iles, Raminta Zmuidinaite, Christoph Sadee, Anna Gardiner, Jonathan Lacey, Stephen Harding, Jernej Ule, Debra Roblett, Jonathan Heeney, Helen Baxendale, Ray K. Iles

**Affiliations:** 1MAPSciences, The iLab, Stannard Way, Bedford MK44 3RZ, UK; jasoniles51@googlemail.com (J.I.); raminta.zmuidinaite@mapsciences.com (R.Z.); anna.gardiner@mapsciences.com (A.G.); jon.lacey@mapsciences.com (J.L.); 2Laboratory of Viral Zoonotics, Department of Veterinary Medicine, University of Cambridge, Madingley Road, Cambridge CB3 0ES, UK; jlh66@cam.ac.uk; 3Francis Crick Institute, Kings Cross, London NW1 1AT, UK; ch.y.sadee@gmail.com (C.S.); jernej.ule@crick.ac.uk (J.U.); debbie.roblett@crick.ac.uk (D.R.); 4The Binding Site Group Ltd., 8 Calthorpe Road, Edgbaston, Birmingham B15 1QT, UK; stephen.harding@bindingsite.com; 5DIOSynVax, University of Cambridge, Madingley Road, Cambridge CB3 0ES, UK; 6Royal Papworth Hospital NHS Foundation Trust, Cambridge CB2 0AY, UK; hbaxendale@nhs.net; 7NISAD, Sundstorget 2, 252-21 Helsingborg, Sweden

**Keywords:** COVID-19, convalescent plasma, spike protein, nucleocapsid, glycated albumin, semi-automated magnetic rack

## Abstract

The immune response to SARS-CoV-2 infection requires antibody recognition of the spike protein. In a study designed to examine the molecular features of anti-spike and anti-nucleocapsid antibodies, patient plasma proteins binding to pre-fusion stabilised complete spike and nucleocapsid proteins were isolated and analysed by matrix-assisted laser desorption ionisation–time of flight (MALDI-ToF) mass spectrometry. Amongst the immunoglobulins, a high affinity for human serum albumin was evident in the anti-spike preparations. Careful mass comparison revealed the preferential capture of advanced glycation end product (AGE) forms of glycated human serum albumin by the pre-fusion spike protein. The ability of bacteria and viruses to surround themselves with serum proteins is a recognised immune evasion and pathogenic process. The preference of SARS-CoV-2 for AGE forms of glycated serum albumin may in part explain the severity and pathology of acute respiratory distress and the bias towards the elderly and those with (pre)diabetic and atherosclerotic/metabolic disease.

## 1. Introduction

The disease syndrome COVID-19 is caused by the enveloped coronavirus SARS-CoV-2. Following infection, its clinical manifestation ranges from very mild, even asymptomatic, to life threatening. Antibodies against SARS-CoV-2 envelope proteins are seen in infected individuals and are particularly elevated in those with severe illness from the infection [1].

The initial infection of the nasopharyngeal epithelia can be asymptomatic, and in many people, particularly the young, the local immune response, such as mucosal immunoglobulin (Ig)A, will be sufficient to prevent the virus from progressing. However, if it does progress to involvement of the upper airways, the disease manifests with symptoms of fever, malaise and dry cough. During this phase, there is a greater immune response involving the release of C-X-C motif chemokine ligand 10 (CXCL-10) and interferons (IFN-β and IFN-λ) from virus-infected cells [2]. The majority of patients do not progress beyond this phase, as the mounted immune response is sufficient to contain the spread of infection. 

About one-fifth of all infected patients progress to involvement of the lower respiratory tract and acute respiratory distress syndrome (ARDS). Virus-laden pneumocytes release many different cytokines and inflammatory markers such as interleukins, tumour necrosis factor-α, interferons, monocyte chemoattractant protein-1 and macrophage inflammatory protein-1α. This ‘COVID-19 cytokine storm’ acts as a chemoattractant for neutrophils, CD4 helper T cells and CD8 cytotoxic T cells. Sequestered into the lung tissue, these cells are responsible for the subsequent inflammation and lung injury, culminating in an acute respiratory distress syndrome [3,4].

At admission, the majority of patients had lymphopenia and platelet abnormalities, elevated neutrophils, aspartate aminotransferase (AST) and lactate dehydrogenase (LDH) along with these inflammatory biomarkers [5]. Medical imaging (CT or X-ray) frequently shows that patients have bilateral pneumonia, and pleural effusion occurs in only 10% of patients [6]. Compared to patients in general, refractory patients have higher levels of neutrophils, AST, LDH and C-reactive protein (CRP) but lower levels of platelets [7,8] and albumin [9,10].

The spike complex is the principal target for neutralising antibodies, as this structure, exposed on the viral envelope surface, is functionally responsible for the virus targeting and entering cells via the angiotensin-converting enzyme (ACE) 2 receptor expressed on target cells. The spike complex is a trimer of the large S protein that, after binding to the ACE-2 receptor, undergoes activation via two-step protease cleavage: the first one at the S1/S2 cleavage site is for S protein priming, and the second cleavage at a position adjacent to a fusion peptide within the S2 subunit is for its activation [11,12,13,14]. The initial cleavage stabilises the S2 subunit at the attachment site, and the subsequent cleavage activates the S protein, causing conformational changes that lead to viral and host cell membrane fusion [15].

As part of an investigation of serum/plasma binding proteins to SARS-CoV-2 antigenic proteins, glycated albumin was found to bind strongly to the stabilised complete spike protein but not nucleocapsid protein nor control protein G. Characterised by MALDI-ToF mass spectrometry, the bound albumin was found to vary in molecular mass, consistent with advanced glycation end products. 

## 2. Results

This study relied on the use of antigen-coupled magnetic beads to capture human serum proteins, which requires robust and reproducible processes of magnetic bead capture, washing, agitation and target-binding protein elution (Figure 1C,D). If this is performed manually, the result can vary dramatically due to variations in the timing between steps and in the intensity of mixing, while insufficient washing can result in the recovery of large amounts of non-specific binding proteins and lead to variable efficiency of target-binding protein recoveries. To overcome these problems and to minimise individual operator variability, we employed a newly developed automated magnetic rack system. By enabling automation, this system leads to consistent timing and intensity of binding, mixing and elution steps and ensures gentle but thorough washing by enabling longer mixing with more repetitions.

After elution from the respective antigen-coupled magnetic beads, MALDI-ToF mass spectra were obtained, and peaks were recorded. These were matched against reference MALDI-ToF mass spectra from the preparation of purified human serum proteins run under the same reducing and acetic acid pH conditions: human serum albumin (HSA), transferrin (Merck Life Science UK Ltd., Dorset, UK), IgG1, IgG3, IgA and IgM (Abcam, Discovery Drive, Cambridge, Biomedical Campus, Cambridge, UK).

Although immunoglobulin light chains and heavy chains were recovered as expected (and subject to detailed analysis reports elsewhere), there was a significant recovery of HSA by the magnetic beads to which stabilised complete spike was conjugated (Figure 2). Indeed, the recovery was conspicuous in this one magnetic bead–coupled SARS-CoV-2 protein series, which recovered levels of HSA similar to that seen for protein G binding and recovery of human IgG1, to which it has a known specificity. The same samples failed to bind albumin to the same degree despite its vast abundance in the samples relative to protein G or nucleocapsid (see Figure 2). Human serum albumin was detected in 72% of the samples of the spike protein–conjugated magnetic beads (compared to 33% for nucleocapsid and 13% protein G) and at an intensity level over 10x higher than that found for residual non-specific binding of HSA for nucleocapsid and protein G (mean values: 1923 AU spike protein, 159 AU N protein and 100 AU protein G).

Closer examination of the bound and eluted HSA masses revealed that the average mass of the albumin recovered by the stabilised complete spike protein was higher than that found for the low-level HSA that non-specifically absorbed to nucleocapsid and protein G (Δ152 *m*/*z*, *p* < 0.05, see Figure 3, panel A). The binding of this higher molecular mass albumin was not a specific unique feature of plasma from patients who had recovered from COVID-19 ARDS but was also found to occur in seronegative and seropositive healthcare worker (HCW) samples (see Figure 3, panel B). The average increase in HSA mass was 152 *m*/*z* but ranged from 50 *m*/*z* up to 500+ *m*/*z*, and spike protein affinity was greatest for albumin with an increased mass of about 150 *m*/*z* (see Figure 3, panel B).

## 3. Discussion

By far, the group at greatest risk of dying following infection with SARS-CoV-2 comprises those over 60 who develop COVID-19 symptoms, accounting for nearly 95% of all such deaths. The next most dominant associations with severe symptoms and COVID-19 mortality are hypertension, obesity and type II diabetes [16,17,18,19]. 

The underlying pathological mechanism of these associations is unknown, but it is likely to be multifactorial rather than a single determinant/marker [20]. 

Strong and specific HSA binding to magnetic beads, to which stabilised prefusion spike protein is conjugated, was found to occur with all plasma samples, be it from seronegative and seropositive medical staff (who had no symptoms or mild symptoms) or from COVID-19 patients who had been treated for ARDS on COVID-19 intensive therapy unit (ITU) wards. Albumin in these same plasma samples was not found to bind strongly to nucleocapsid- or protein-G-coated magnetic beads, which acted as a control. This is despite the fact that significant human serum albumin binding has been reported as a feature of protein G [21]. 

Many microorganisms that cause seriously harmful infections have developed mechanisms to evade the immune system. One such mechanism includes the specific binding of serum proteins to mask antigenic sites and thus evade antibody neutralisation and other immune processes. The spike protein is an extremely large antigenic target projecting from and exposed on the virion envelope. It is the focus of neutralising antibodies [22]. Thus, the ability of the virus to evade neutralising antibodies by coating itself with abundant serum protein(s), until such time as target cell receptor binding has occurred, is an ideal strategy to evolve, particularly when entering the bloodstream. Bacteria express families of genes coding for surface proteins that bind serum proteins, particularly those that bind immunoglobulins, i.e., IgG by protein G (human group C and G streptococci) and protein A (*Staphylococcus aureus).* An analogous gene to that which codes for protein G has been found to code for a protein termed Peptostreptococcal albumin-binding protein (PAB), which binds human serum albumin. PAB is highly expressed by the anaerobic bacterium *Peptostreptococcus magnus* [21], and such expression has been correlated with the virulence of *P. magnus* strains. 

Similarly, the viral nanoparticle surface can absorb host proteins as a coat or protective corona, and this in turn affects the host’s physiology and immune response [23,24,25]. For example, respiratory syncytial virus (RSV) and herpes simplex virus type 1 (HSV-1) accumulate a rich protein corona from biological fluids, and this protein corona affects viral infectivity and immune cell activation [25]. Recent molecular modelling has indicated that the S1 subunit of the spike protein would bind HSA in such a way that it blocks access to antigenic sites on the receptor-binding domain (RBD) [26]. Thus, even weak binding affinity for HSA could provide a virulence enhancement via some protection/evasion from RBD neutralisation antibodies. 

Since the eluted bound albumin was detected via MALDI-ToF mass spectrometry, the molecular mass of the bound albumin could be determined with greater precision. It was found that the molecular mass of the elevated levels of albumin binding to the stabilised spike protein was higher than expected and higher than that of the low levels of non-specific bound albumin, which could be detected in some elution samples from nucleocapsid- and protein-G-coated magnetic beads (Figure 3). 

Viruses, such as influenza, bind to specific glycans as part of their attachment and invasion mechanisms; e.g., the viral envelope protein haemagglutinin recognises and binds sialic acid glycan residues [27]. Albumin is not a glycosylated protein [28]; however, the multiple increases in molecular mass seen here correspond to known glycation products termed AGEs, which form from blood saccharide reactions with long-lived and abundant serum proteins. The AGE reaction occurs on specific lysine and arginine amino acid side chains of HSA (see Figure 4). Glycation of serum proteins is more rapid and abundant than glycation of haemoglobin and has been used as a biomarker for both diabetes and arthrosclerosis [29]. 

Paradela-Dobarra et al. [30] demonstrated that AGE modification of albumin could be detected by direct MALDI-ToF mass spectrometry via an increase in the averaged albumin peak molecular mass. This was reflected in a ratio-consistent increase in mass for the singly and doubly charged peaks of albumin. We also detected a similarly increased average mass of singly and doubly charged HSA peaks, determined by MALDI-ToF MS, in our samples, and the results are consistent with AGE glycation forms of HSA being preferentially captured by the spike protein. 

A stronger affinity to glycated/AGE forms of serum albumin would enhance such a virulence effect of coronavirus in those with pre-diabetes and metabolic diseases.

## 4. Materials and Methods

### 4.1. Samples

Serum and plasma samples were obtained from HCWs and COVID-19 patients referred to the Royal Papworth Hospital for critical care during the first wave. NHS healthcare workers working at the Royal Papworth Hospital in Cambridge, UK, served as the exposed HCW cohort (study approved by Research Ethics Committee Wales, IRAS: 96,194 12/WA/0148. Amendment 5). NHS HCW participants from the Royal Papworth Hospital were recruited through staff email over the course of two months (20th April 2020–10th June 2020) as part of a prospective study to establish seroprevalence and immune correlates of protective immunity to SARS-CoV-2. Patients in convalescence were recruited either pre-discharge or at the first post-discharge clinical review. All participants provided written informed consent prior to enrolment in the study. Sera from NHS HCWs and patients were collected between July and September 2020, approximately three months after they were enrolled in the study [31]. 

Clinical assessment and WHO criteria scoring of severity for both patients and NHS HCWs were conducted following ‘*COVID-19 Clinical Management: living guidance*’ (https://www.who.int/publications/i/item/WHO-2019-nCoV-clinical-2021-1, accessed 14 June 2021). 

For cross-sectional comparison, representative convalescent serum and plasma samples from seronegative HCWs, seropositive HCWs and convalescent polymerase chain reaction (PCR)-positive COVID-19 patients were obtained. Serological screening to classify convalescent HCWs as positive or negative was carried out according to the results provided by a CE-validated Luminex assay detecting N-, RBD- and S-specific IgG, a lateral flow diagnostic test (IgG/IgM) and an electro-chemiluminescence immunoassay (ECLIA) detecting N- and S-specific IgG. Any sample that produced a positive result in any of these assays was classified as positive. The clinical signs of the individual from whom the sample was obtained ranged from 0 to 7 according to the WHO classification described above. Thus, the convalescent plasma samples (three months post-infection) were grouped into three categories: (A) Seronegative Staff (*n* = 30 samples); (B) Seropositive Staff (*n* = 31 samples); (C) Patients (*n* = 38 samples).

### 4.2. Antigen-Coupled Magnetic Beads

The viral spike protein (S protein) is present on virions as a pre-fusion trimer with the receptor-binding domain of the stochastically open or closed S1 region, an intermediary where the S1 region is cleaved and discarded, and S2 undergoes major conformational changes to expose and then retract its fusion peptide domain [32]. Here, the S protein was modified to disable the S1/S2 cleavage site and maintain the pre-fusion stochastic conformation [33]. Protein-G-coupled magnetic beads were purchased from Cytivia Ltd. (Amersham Place, Little Chalfont, Buckinghamshire, UK). Recombinant nucleocapsid and recombinant stabilised complete spike protein magnetic beads were made by Bindingsite Ltd. (Birmingham, UK). 

### 4.3. Isolation of Bound Proteins with MagMix, a Semi-Automated Magnetic Rack

Consistency in the processing of magnetic bead–coupled antigen-captured binding proteins from plasma samples was achieved using the semi-automated magnetic rack produced by the Crick Institute, London, UK (see Figure 1, panel 1; further details at bitomix.com accessed 14 June 2021). This instrument has a series of magnets on rotating discs set on either side of the tube containing samples and magnetic beads. Rotation of the magnets not only pulls the antigen-coupled magnetic beads to one side, as is implemented by commercially available static magnets, but also has a series of other automated functions. These include:

1. Engage/Disengage: Beads are immobilised, allowing liquid to be aspirated or disengaged to transfer beads in solution to a new test tube.

2. Mix: Beads are pulled from one side of the tube to the other under a changing magnetic field in a gentle but thorough manner, enabling repetitive and consistent washing.

3. Resuspend: Beads are suspended in liquid via a rapidly changing magnetic field and remain in liquid when magnets disengage after three cycles, allowing beads to be further processed or transferred to a new tube.

4. Remote: Beads are washed and agitated via a custom protocol to be uploaded through the serial port.

To prepare the beads, empty 1.5 μL microcentrifuge tubes were loaded into the semi-automated magnetic rack. The rack was turned on and set to the ‘Disengage’ setting (Figure 1C). Protein G (GE), purified nucleocapsid or purified stabilised complete spike magnetic beads (Bindingsite, Birmingham, UK) in their buffer solutions were vortexed to ensure an even distribution of beads within the solution. First, 10 μL of the appropriate magnetic beads was pipetted into each tube. Then, 100 μL of wash buffer, 0.1% Tween 20 in Dulbecco’s phosphate-buffered saline (DPBS), was added to each tube before the rack was set to the ‘Resuspend’ setting. During this period, the magnets continuously circulated for several rotations before stopping. Once the rack had finished resuspending, the settings were changed to ‘Mix’ for several minutes and then changed back to the ‘Resuspend’ setting to mix once more. The rack was set to ‘Engage’, allowing the wash buffer to be carefully discarded using a pipette while taking care not to disturb the beads. The wash cycle was repeated 3 times. After 3 wash cycles, all of the remaining wash buffer in the tubes was discarded without disturbing the beads. The rack was set to the ‘Disengage’ setting.

To process the samples and elute the binding fraction from magnetic beads, 45 μL of 10x DPBS was pipetted into each of the tubes containing the washed magnetic beads. Then, 5 μL of vortexed neat plasma was pipetted and pump-mixed into a tube containing the beads, repeating this step for each plasma sample. The rack was set to the ‘Resuspend’ setting to ensure thorough mixing of the beads with the now diluted plasma. After the resuspension, the rack was set to ‘Mix’ for 20 min. After mixing, the rack was set to ‘Engage’, and the remaining solution was carefully discarded so as to not disturb the beads. A further 3 wash cycles were conducted using 0.1% Tween 20 in DPBS. Subsequently, another 3 wash cycles were conducted using ultra-pure water, discarding the water after the last cycle, and 15 μL of recovery solution (20 mM tris(2-carboxyethyl)phosphine (TCEP) (Sigma-Aldrich, UK) + 5% acetic acid + ultra-pure water) was pipetted into the tubes. The tubes were removed from the rack and mixed by flicking the tube to fully disturb the beads. All of the liquid was sent to the bottom of the tube by firmly tapping the tube on a surface to bring the droplets down. The tubes were securely placed back into the rack and were run alternatively between the ‘Resuspend’ setting and the ‘Mix’ setting for 5 min. The rack was set to ‘Engage’, and the recovery solution was carefully removed using a pipette and placed into a clean, labelled 0.6 μL microcentrifuge tube. This recovery solution was the eluant from the beads and contained the desired proteins.

### 4.4. Sample Analysis by MALDI-ToF Mass Spectrometry

Mass spectra were generated using a 15 mg/mL concentration of sinapinic acid (SA) matrix. The eluate from the beads was used to plate with no further processing, and 1 µL of the eluted samples was taken and plated on a 96-well stainless-steel target plate using a sandwich technique. The MALDI-ToF mass spectrometer (microflex^®^ LT/SH, Bruker, Coventry, UK) was calibrated using a 2-point calibration of 2 mg/mL bovine serum albumin (33,200 *m*/*z* and 66,400 *m*/*z*) (Pierce^TM^, ThermoFisher Scientific). Mass spectral data were generated in positive linear mode. The laser power was set at 65%, and spectra were generated at a mass range between 10,000 and 200,000 *m*/*z*; pulsed extraction was set to 1400 ns. 

A square raster pattern consisting of 15 shots and 500 profiles per sample was used to obtain 7500 total profiles per sample. The average of these profiles was generated for each sample, resulting in a reliable and accurate representation of the sample across the well. The averaged raw spectral data were then exported in a text file format to undergo further mathematical analysis.

### 4.5. Spectral Data Processing

Mass spectral data generated by the MALDI-ToF instrument were uploaded to the open-source mass spectrometry analysis software mMass™ [34], where it was processed by using a single-cycle Gaussian smoothing method with a window size of 300 *m*/*z* and baseline correction with applicable precision and relative offset, depending on the baseline of each individual spectrum. In the software, automated peak-picking was applied to produce peak lists, which were then tabulated and used in subsequent statistical analysis. 

### 4.6. Statistical Analysis

Peak mass and peak intensities were tabulated in Excel and plotted in graphic comparisons of distributions for each antigen capture and patient sample group. Means and medians were calculated, and, given the asymmetric distributions found, non-parametric statistics were applied, such as the Mann–Whitney U test, when comparing differences in group distributions.

## 5. Conclusions

An unusual feature of the COVID-19 disease is microthrombosis and localised disruption of the osmotic potential with pulmonary microvascular dilation. Pulmonary thrombosis is common in sepsis-induced ARDS. Coagulation dysfunction appears to be common in COVID-19 and is detected by elevated D-dimer levels. In fatal cases, there is diffuse microvascular thrombosis, suggesting thrombotic microangiopathy, and most deaths from COVID-19 ARDS have evidence of thrombotic disseminated intravascular coagulation [35]. This may explain some of the atypical or unexpected manifestations seen in the lung, such as dilated pulmonary vessels on a chest computerised tomography (CT) scan and episodes of pleuritic pain. Vascular enlargement is rarely reported in typical ARDS yet has been seen in most cases of COVID-19 ARDS [36].

Low circulating serum albumin is associated with pleural effusion, oedema and vascular constriction as a result of serum albumin being sequestered within the interstitial tissues. However, in COVID-19 ARDS, we see a lower overall serum albumin level, and yet pulmonary vascular dilation still occurs. Thus, where is the serum albumin being lost to? Using intratracheally instilled SARS-CoV-2 spike protein, it was demonstrated in a mouse model that total protein concentration in bronchoalveolar lavage fluid (BALF) dramatically increased [37].

At this point, it is not feasible for us to perform a parallel analysis and comparison of the albumin-binding S protein of serum with corresponding BALF samples. However, it is an important direction for prospective studies for researchers who have access to both sample types in order to better understand where serum albumin is lost.

The binding of albumin by SARS-CoV-2 has been postulated by Johnson et al. [38] as a molecular contributor to the tissue–vascular fluid imbalance that gives rise to septic shock in COVID-19 cases. It is possible that it is being caught locally within the pulmonary vasculature and contributing to the pathologic mechanism of local vascular dilation, e.g., within microthrombolytic clots found in the pulmonary blood vessels [39,40]. Furthermore, deposition of glycated/AGE-modified albumin has also been correlated with the hyperinflammatory process seen in cardiovascular disease [34].

In conclusion, the stabilised/prefusion spike protein has a binding affinity for human serum albumin, particularly glycated/AGE forms of albumin. The correlation of type 2 diabetes, obesity and age with a greater probability of suffering from ARDS as a result of SARS-CoV-2 infection that persists and progresses may in part be due to the higher affinity of the spike complex for glycated/AGE forms of serum albumin. The ability to evade effective immune responses by hiding behind a coating of serum proteins is much like an octopus grabbing shells and stones to hide from predatory sharks (*Blue Planet II* 2017) [41], but in doing so, it may cause the tissue–vascular fluid physiological changes seen in COVID-19 ARDS, even in those with subclinical pre-diabetes and metabolic diseases.

## 6. Patents

No patents have been filed as a result of this study.

## Figures and Tables

**Figure 1 ijms-23-04126-f001:**
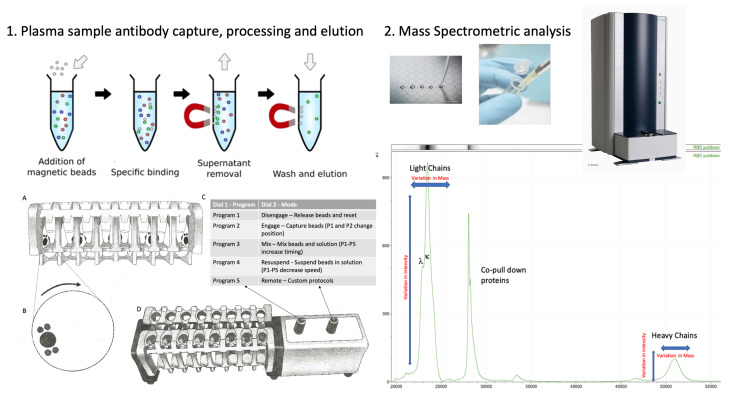
A schematic of the capture and elution of binding proteins, such as immunoglobulins, is illustrated in panel **1**. Plasma samples are mixed with magnetic beads coupled with nucleocapsid (N protein), stabilised complete spike (spike protein) or G protein. The magnetic particle processing rack can simultaneously process 16 samples in Eppendorf tubes (**A**). Using rotating discs containing magnets (**B**) on either side of the sample allows capture and removal of denuded sample, mixing and efficient wash off of non-specific absorption and final elution of the capture binder proteins (**C**,**D**) in a consistent standardised process. The eluted samples are spotted onto a MALDI-ToF mass spectrometry plate along with MALDI matrix and subjected to mass spectral analysis following a standardised protocol and settings. Raw data are exported and analysed to give precise measurements of molecular masses of the eluted binder/complex proteins (e.g., immunoglobulin light and heavy chains), along with intensity measurement for relative quantification purposes (panel **2**).

**Figure 2 ijms-23-04126-f002:**
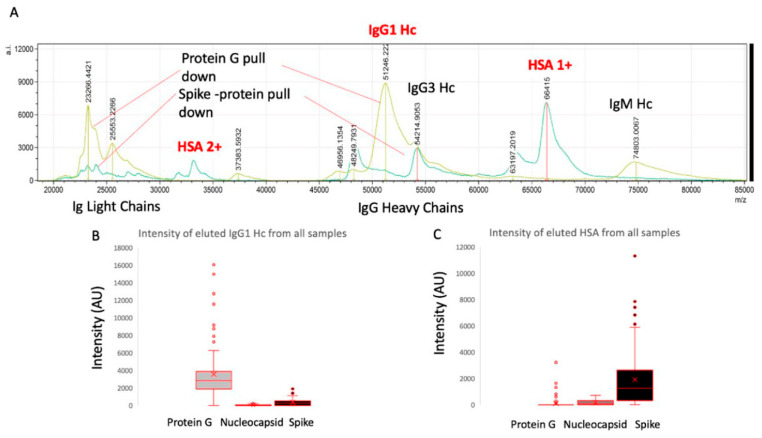
Example mass spectra of eluted sample from protein G bead–captured proteins overlaid with spike protein–captured plasma proteins (panel **A**) and relative intensities of IgG1 heavy chains (IgG 1 Hc) recovered from the same samples by protein G, nucleocapsid and stabilised spike protein (panel **B**) versus HSA (single: 1+; doubly charged: 2+) recovered from the same samples by protein G, nucleocapsid and stabilised spike protein (panel **C**).

**Figure 3 ijms-23-04126-f003:**
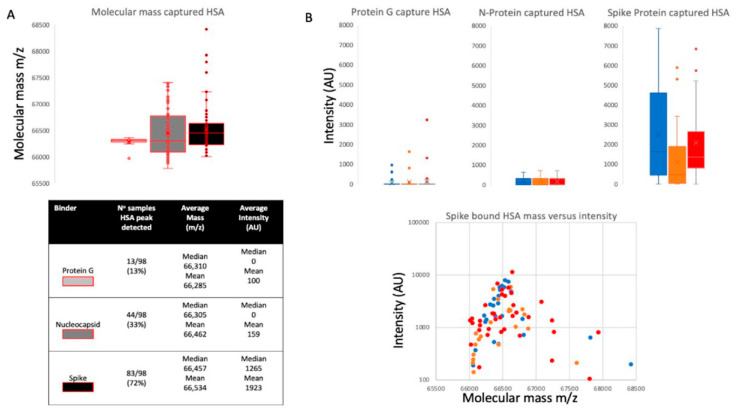
Distribution of mass determined for captured and eluted human serum albumin (HSA) and the relative intensity of the increased mass of bound albumins: Panel **A** illustrates the mass variance of protein G, nucleocapsid (N protein) and stabilised complete spike protein along with average intensity. Panel **B** illustrates a lack of differences in HSA binding by protein G, nucleocapsid and stabilised complete spike protein among the patient group’s plasma samples, whilst there is a clear preference of the S protein to bind higher molecular weight albumin in all samples. Blue represents data from SARS-CoV-2 seronegative HCWs, orange represents data from SARS-CoV-2 seropositive HCWs who have recovered from the infection with mild symptoms, and red represents sample data from convalescent patients recovering from COVID-19 ARDS.

**Figure 4 ijms-23-04126-f004:**
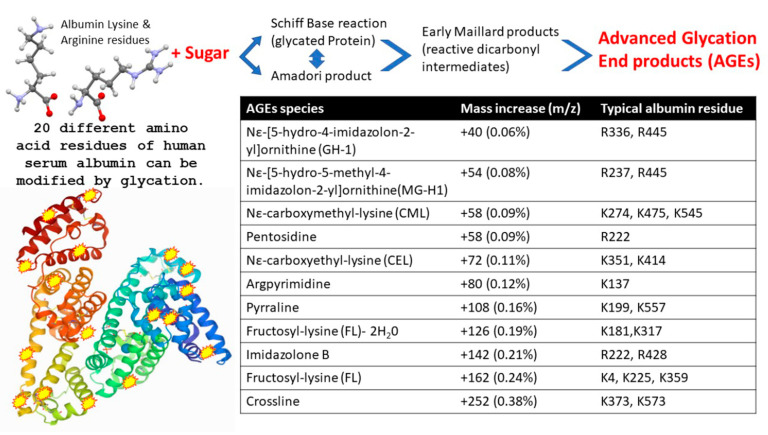
Illustration of advanced glycation end product (AGE) as a result of human serum albumin Schiff base/Amadori/Maillard reactions with blood circulating reducing monosaccharides (glucose and fructose), resulting in modification to exposed lysine and arginine amino acid side chains. The molecular mass effects of these AGE reactions on specific HSA residues are also detailed.

## Data Availability

Compiled summary data can be made available upon request to the corresponding author. Raw mass spectral data from the individual samples will require compiling from archives at MAPSciences and so require a detailed project proposal to justify the additional resource expenditure required in providing this complete data set.

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
