# Peer review of "SARS-CoV-2 Spike Protein Binding of Glycated Serum Albumin—Its Potential Role in the Pathogenesis of the COVID-19 Clinical Syndromes and Bias towards Individuals with Pre-Diabetes/Type 2 Diabetes and Metabolic Diseases"

_ijms, 2022, doi:10.3390/ijms23084126_

Round 1

Reviewer 1 Report

Iles et al. revealed a preferential capture of advanced glycation end products (AGE) glycated human serum albumin by the pre-fusion spike protein by mass spectroscopy. This is good work in the current perspectives. Authors also reported that the preference of SARS-CoV-2 for AGE glycated serum albumin may in part explain the severity and pathology of acute respiratory distress and a bias to the elderly, and those with (pre)diabetic and atherosclerotic/metabolic. Do you have experimental proof or cross-sectional study to link the interrelation? If not, do you consider the title and bias towards individuals with pre-diabetes/type 2 diabetes & metabolic diseases”.

Author Response

We would like to thank reviewer for their comments and suggestions. We have acknowledged the comments for each reviewer and addressed each question/suggestion line by line as follows:

There are two subsequent papers in the series that further explore the link and detail the published literature on the bias towards individual with type 2 diabetes and metabolic disease. In addition we have included supplemental tables which details the demographic features and co-morbidities of the study patients. Thus, we would like to leave the title as initially proposed.

Reviewer 2 Report

The manuscript is well written. The findings are novel and will be interesting for readers. However, there are some concerns about the questions asked and the answers below in the findings. 

The authors ask, where is the serum albumin being lost? It is well known that, in addition to interstitial effusion, a large amount of serum protein is shed into the alveolar space, which is associated with endothelial barrier dysfunction. This has been clearly demonstrated in a mouse model using the SARS-CoV-2 spike protein Subunit 1 (Colunga Biancatelli et al, 2001). The analysis of bronchoalveolar lavage fluid S protein binding of glycated serum albumin of COVID-19 patients would be very useful for this study along with studied serum albumin. However, It's understandable that it's impossible to take BAL samples from the same patients anymore. Authors should discuss their interesting theory in the context of bronchoalveolar lavage albumin and future possibilities to perform this study and modify conclusions.

Author Response

Thank you for your comments and suggestions.
As the reviewer suggested, we are unable to perform the comparative analysis on a bronchoalveolar lavage fluid from the same patients.
The paragraph and appropriate reference was added to the discussion, lines 350 to 356. We have also added supplemental tables concerning demographics of the study cohorts and patient co-morbidities.

Round 2

Reviewer 2 Report

The manuscript could be accepted in present form